# Understanding SNAP Recipient Characteristics to Guide Equitable Expansion of Nutrition Incentive Programs in Diverse Food Retail Settings

**DOI:** 10.3390/ijerph19094977

**Published:** 2022-04-20

**Authors:** Lauren Vargo, Timothy H. Ciesielski, Milen Embaye, Ana Bird, Darcy A. Freedman

**Affiliations:** 1Mary Ann Swetland Center for Environmental Health Research, Department of Population and Quantitative Health Sciences, Case Western Reserve University School of Medicine, Cleveland, OH 44106, USA; lnv10@case.edu (L.V.); thc23@case.edu (T.H.C.); mxe198@case.edu (M.E.); 2Produce Perks Midwest, Cincinnati, OH 45241, USA; ana@produceperks.org

**Keywords:** nutrition incentives, supplemental nutrition assistance program, health disparities, health equity, food security, food purchasing

## Abstract

Structural barriers, such as food costs, reduce access to healthy foods for populations with limited income, including those benefitting from the Supplemental Nutrition Assistance Program (SNAP). Nutrition incentive programs seek to address this barrier. Evaluations of SNAP-based incentive programming often focus on one setting (i.e., either farmers’ markets or grocery stores). We examined use patterns, characteristics, and preferences among 253 SNAP consumers with access to incentive programming at both a farmers’ market and a grocery store located within five miles of their home. Cross-sectional survey data were collected in 2019 in two Ohio cities. Despite geographic access, 45% of those surveyed were not using the incentive program; most non-users (80.5%) were unaware of the program. Program users compared to non-users had higher household incomes (*p* < 0.001) and knew more people using the program (*p* < 0.001). Grocery stores were the most common setting of use (59%); 29% used at farmers’ markets; 11% used in both settings. User characteristics varied by store setting based on demographics, program experience, fruit and vegetable purchasing and consumption patterns, and social dynamics related to use. Our findings support comprehensive awareness-raising efforts and tailored implementation of incentive programming that attends to diverse segments of SNAP consumers to promote equity in program reach.

## 1. Introduction

Food insecurity is a persistent and pressing public health inequity in the U.S., with 10.5% of households affected nationally in 2019; trends did not improve throughout 2020 during the COVID-19 pandemic when food assistance benefits were increased [1,2]. The Supplemental Nutrition Assistance Program (SNAP) is a federal food assistance program that provides income-eligible individuals and families with monthly benefits to purchase food, aiming to supplement limited food budgets and improve food security [1,3]. Even with these financial supplements, people receiving SNAP face structural barriers to achieving a healthy diet, citing the lack of affordability of healthy foods as the top barrier [4]. While SNAP is estimated to reduce the probability of being food insecure among recipients [5,6], recipients report having poorer diet quality and higher rates of diet-related diseases when compared to income-eligible non-recipients [7,8]. People receiving SNAP also report periods of food insecurity and a decline in diet quality throughout the month as financial resources are spent [7] and often have nutrient deficiencies that can be directly addressed through increased intake of fruits and vegetables [9].

The Food Insecurity Nutrition Incentive (FINI) Program, now known as the Gus Schumacher Nutrition Incentive Program (GusNIP), was established in the U.S. in the 2014 Farm Bill to provide financial incentives to SNAP recipients for purchasing fruits and vegetables [10]. Evaluations of SNAP-based nutrition incentive programs found they increase spending on fruits and vegetables and are associated with reductions in food insecurity and improvements in diet quality [11,12]. However, the reach of the program is limited in many communities; more implementation research is needed to increase the number of people who are benefiting from the program [13].

Ohio is a rich context for examining implementation of SNAP-based nutrition incentive programming. Ohio’s nutrition incentive program, Produce Perks, has evolved over the past decade from single-site operations at farmers’ markets into a statewide, networked, multi-food retail setting program [14]. Like the approach in Ohio, national nutrition incentive programming initially focused on farmers’ markets. However, evaluation of the Healthy Incentives Pilot (HIP) in 2012 found that less than 1% of SNAP households shopped at farmers’ markets during any given month [15], a trend that persisted as of 2019 [16]. In contrast, about 40% of SNAP benefits are spent at a supermarket or grocery store [16]. Accordingly, there are growing efforts to diversify and expand program operations in grocery stores across the U.S. [11,12]. This allows implementation to occur in multiple food retail settings within a community, promoting choice and nutrition equity among SNAP recipients [17].

Most extant research is focused on the potential impacts of incentive program implementation at either farmers’ markets [18,19,20,21] or grocery stores [11,12]. Little is known about factors influencing decisions to use SNAP-based incentive programs when they are available at multiple food retail settings in one community, such as a farmers’ market and a grocery store. In this research, we address this gap by taking a holistic view to examine use patterns, characteristics, and preferences among SNAP recipients with access to nutrition incentive programming at both a farmers’ market and a grocery store located within five miles of their home. The study is guided by an established nutritious food access framework allowing for examination of economic, spatial–temporal, service delivery, social, and personal factors that may shape incentive program use [22]. Overall, our goal was to inform future research and interventions that promote equitable program access in similar urban contexts.

## 2. Materials and Methods

### 2.1. Study Description

Our study was conducted by a collaborative team that included an academic research center and a statewide non-profit organization that supports implementation of Ohio’s nutrition incentive program. We performed a cross-sectional survey from July–October 2019 in two cities in Ohio: Akron and Cleveland. In 2019, the populations of Akron and Cleveland were about 198,000 and 385,000, respectively, and both cities resided in counties that had over 13% of persons living in poverty [23]. These cities met study eligibility requirements, including: (1) having a grocery store operating and offering Produce Perks year-round for a minimum of six months prior to data collection, (2) having a farmers’ market operating and offering Produce Perks year-round for a minimum of six months prior to data collection, and (3) the grocery store and farmers’ market being located within three miles of one another. Our goal was to recruit participants with comparable and reasonable geospatial access to Produce Perks in both retail settings in their community to allow for examination of other potential access barriers per the nutritious food access framework [22]. Accordingly, we drew a buffer around the centroid of both the farmers’ markets and grocery store in each city using ArcGIS to represent the Euclidean distance of five miles [24]. We recruited people receiving SNAP who lived within this five-mile buffer. In 2015, we observed that SNAP recipients in Cleveland traveled a median distance of 2.3 miles when they shopped at a farmers’ market [25]. Thus, by choosing sites that were no more than 3 miles apart and drawing 5-mile eligibility circles around these sites, we ensured that travel distance to either option was reasonable. The Case Western Reserve University Institutional Review Board reviewed and approved the study protocol (20190723).

### 2.2. Study Participants

We conducted participant recruitment by partnering with the statewide non-profit organization that manages nutrition incentive programming and the statewide social services agency that operates SNAP benefits. The research team worked with these partners to distribute two targeted mailers to SNAP recipients within the study boundaries in both cities. The mailers sent by the county-based social service agencies included: (1) a promotional flyer about Produce Perks with the location and details of a grocery store and year-round farmers’ market located near their home, information about how to use the incentive program, and resources to find more information about Produce Perks; and (2) a study recruitment flyer listing basic eligibility criteria and information for the SNAP recipient to contact the research team if interested in learning more about the study. Additional recruitment efforts included in-person tabling at all four promoted farmers’ market and grocery store sites throughout the data collection period and distribution of recruitment flyers to local organizations, community events, apartment buildings, and food pantries.

We used a six-item screener to determine eligibility for this cross-sectional survey. Screener questions were asked by a trained research assistant in person or over the phone. Individuals were eligible if they were: (1) currently receiving SNAP benefits; (2) residing within the defined study boundaries; (3) responsible for at least half of the food shopping for their household; (4) ≥18 years of age; (5) English-speaking; and (6) the only member of their household enrolled in the study. To determine whether a potential participant resided within the defined study boundaries, they were asked to provide their home address, which was then entered into ArcGIS to confirm that it was within the study boundaries. If a participant did not want to provide an address or the address provided was outside of the study boundaries, they were not eligible for the study. Those who were eligible were offered more information on the study, as well as an opportunity to provide consent. We screened 412 people for eligibility, of whom 145 (35.2%) were screened at the in-person events and 267 (64.8%) were screened by phone (Figure 1). Most of those screened were eligible for the study (99.3% of those screened in person and 91.4% of those screened on the phone). We invited 388 people who were eligible to participate and 253 completed the survey (65.2%). About one-third of those eligible did not complete the survey, whether they were screened in person (32.6%) or by phone (36.1%). Prior to completing the survey, study participants provided informed consent in writing (if in person) or verbally (if over the phone).

### 2.3. Data Collection

We developed a structured survey guide for this study (full survey available in Appendix A). Food security was measured using the USDA’s U.S. Household Food Security Survey Module: Six-Item Short Form [26]. Items from the Behavioral Risk Factor Surveillance System (BRFSS) Questionnaire were used to assess health status, fruit and vegetable intake, and demographics [27]. All other questions were derived from the United States Department of Agriculture (USDA) Economic Research Service Current Population Survey (CPS) Food Security Questionnaire [28] or previously conducted studies [27,29]. Overall, the survey contained questions that assessed five domains of nutritious food access: economic, spatial temporal, service delivery, social, and personal [22]. The full list of variables is enumerated in Appendix A and we list some here to explain the approach. In terms of Economic variables, we collected information on annual household income, the amount of monthly SNAP allowance, and the number of people supported by that SNAP allowance. Spatial Temporal variables included: distance to the preferred food shopping store, transportation options/strategies, and frequency of shopping. The Service variables focused on the quality and variety of produce offerings at grocery stores and farmers’ markets, while the Social variables quantified the frequency of Produce Perks and farmers’ market use among friends and family of the participants. Finally, the Personal variables assessed several individual characteristics, perceptions, and skills, including education level, food security, home ownership, frequency of produce consumption, and confidence preparing fruits and vegetables.

All surveys were conducted by phone and were administered by seven trained research assistants. Data were entered directly into REDCap as the survey was being conducted. Surveys took about 30 min. Participants were compensated with a $20 gift card for their time.

### 2.4. Data Analysis

Our primary analytic approach was designed to identify the factors that differ with respect to Produce Perks use status and use location. Program users were compared to non-users, and then users were compared by the setting of their use (grocery store only, farmers’ market only, or both locations). Variables that differed by Produce Perks use status were identified by the Wilcoxon–Mann–Whitney test, the chi-squared test, or Fisher’s exact test as appropriate. Variables that differed by location of use were identified by the Kruskal–Wallis test or Fisher’s exact test as appropriate. The non-parametric tests (Wilcoxon–Mann–Whitney, Fisher’s exact, and Kruskal–Wallis) were used when the assumptions needed for parametric tests were unreasonable to make (e.g., normal distribution or sufficient cell size). Statistical significance was set at *p* < 0.1 to reflect the importance of type II errors in this exploratory research setting. In other words, this is hypothesis-generating exploratory research, and in this setting the learning cost of false negatives outweighs the cost of false positives [30]. The variables analyzed here were organized according to the five domains of the Multi-Component Nutritious Food Access (NFA) Framework [22], as described above (i.e., economic, service delivery, spatial–temporal, social, and personal). The distance between a participants’ reported home address and preferred store address was calculated using Google Maps. Statistical analyses were conducted using SAS 9.4 (SAS Institute; Cary, NC, USA).

## 3. Results

### 3.1. Demographics of Study Participants

A total of 253 SNAP recipients completed the cross-sectional survey (Table 1). Most participants self-identified as female (81.4%), Black (68.4%), and between the ages of 40 and 79 years old (76.6%). While all participants were currently receiving SNAP, the majority (65.6%) reported receiving SNAP for at least five years. Transportation for food shopping varied among participants, with 46.6% driving their own car and 53.4% relying on other options for transportation. On self-report, 30.8% of the participants described their general health status as “fair,” while 8.3% said their health was “poor.” Just over half (55.3%) reported using the Produce Perks nutrition incentive program at least once. Among those, most reported use at a grocery store (59.3%) compared to a farmers’ market (29.3%) or use at both sites (10.7%) (see Figure 1). A full list of descriptive responses is presented in Appendix A.

### 3.2. Rationale for Program Use

When asked why they did not use Produce Perks, 80.5% of the non-users reported that they did not know the incentive program existed. Less frequently cited reasons for not using the program included: confusion about how the program works (8.8%), inconvenient store location (6.2%), transportation hurdles (3.5%), and high food prices at participating stores (2.7%). No participants indicated that their lack of incentive program use was due to a distaste for fruits and vegetables or that their store lacked the produce they wanted. Among the 90 people who did not know about the program until they participated in the study, 76.7% said that they were very likely to use the program in the next 6 months.

The 140 users were asked what motivated them to use the Produce Perks program. The most common motivations for using the incentive program included: the program helps you save money (74%), it allows you to buy more fruits and vegetables (61%), and it is easy to use (14%).

### 3.3. Produce Perks User Profiles

Multiple variables across four of the five domains of access differed between Produce Perks users and non-users at *p* < 0.1 (Table 2): economic, spatial–temporal, social, and personal. In terms of economic factors, Produce Perks users had higher median incomes than non-users (*p* < 0.001). Produce Perks users supported more people with their SNAP (*p* = 0.005) and received more monthly SNAP benefits (*p* = 0.039). With respect to spatial temporal factors, users lived closer to their preferred store (*p* = 0.020) and perceived purchasing fruits and vegetables as more convenient (*p* = 0.005) than non-users (i.e., a lower score reflects higher perceived convenience). Produce Perks users were also more likely to have purchased or received food from a farmers’ market (*p* = 0.034), specialty store (*p* = 0.038), or mobile pantry/free fruit and vegetable drop-off (*p* = 0.044) in the past year. Additionally, Produce Perks users were more likely to have a car for shopping (*p* = 0.051) and were less likely to use somebody else’s car for shopping (*p* = 0.048). Regarding social factors, users reported that more of their friends or family had used Produce Perks (*p* < 0.001) and had shopped at a farmers’ market (*p* = 0.054) in the last 6 months. In terms of personal factors, Produce Perks users had more children in their household (*p* = 0.034). though the median number of children across both groups was less than one. They were also less likely to have moved in the last 12 months (*p* = 0.050) and had higher educational attainment (*p* = 0.061). Finally, when food security was analyzed as a dichotomous variable (i.e., a six-point USDA food security score >1, indicating low or very low food security), Produce Perks users were less likely to report experience of food insecurity (*p* = 0.044) in the past year. Overall, the strongest statistical findings and likely predictors of Produce Perks use were having a higher income and knowing more people who use the incentive program.

When Produce Perks users were compared by their setting of use, several variables differed between the three Produce Perks use types at *p* < 0.1 (Table 3): grocery store only, farmers’ market only, and both. Those who used Produce Perks at a grocery store only (n = 83) had been using the program for the shortest amount of time (*p* < 0.001) and had the lowest education levels (*p* = 0.025). This group reported purchasing fruits and vegetables less frequently (*p* < 0.001) and they had less confidence in their ability to prepare fresh greens (*p* = 0.008) as well as root vegetables (*p* = 0.018). Additionally, grocery store-only Produce Perks users made more frequent trips to their primary store (*p* = 0.009), knew of more friends and family shopping at the same grocery store (*p* = 0.054), and had fewer adults in their household (*p* = 0.067). They also reported that they ate beans (*p* = 0.062) and other vegetables (i.e., beyond beans, dark green or orange vegetables) less frequently (*p* = 0.007). Finally, the grocery store-only users reported a similar variety of fruits and vegetables at their primary store compared to the farmers’ market, while the two groups who utilized farmers’ markets reported a greater variety at their farmers’ markets compared to their primary store (*p* < 0.001). Those using Produce Perks at farmers’ markets only (n = 41) were older (*p* = 0.033), had used Produce Perks for the longest amount of time (*p* < 0.001), and were also less likely to be Black (*p* = 0.056). The group who used Produce Perks at both retail stores and farmers’ markets was the smallest group (n = 15). Users of Produce Perks at both grocery stores and farmers’ markets were redeeming the largest amounts of Produce Perks incentive money (*p* = 0.030). This group was the youngest (*p* = 0.038), most educated (*p* = 0.025), and had the highest levels of confidence in preparing fresh green vegetables (*p* = 0.008). There were no significant differences in food security across the three user types, when specifying food security as a dichotomous variable (*p* = 0.703).

## 4. Discussion

The results of this cross-sectional research add to the field by offering a holistic view of nutrition incentive use and non-use among people receiving SNAP with access to programming in both farmers’ market and grocery store settings near their residence. Our results offer insights that could be used to expand the dietary and food security benefits of nutrition incentive programming. Three key lessons learned from our research include: (1) insufficient awareness of nutrition incentive programming was a critical yet modifiable impediment to its use; (2) multi-setting approaches are needed to engage diverse audiences of SNAP recipients in nutrition incentive programming; and (3) synergistic strategies are needed to reach those who may benefit the most from nutrition incentives.

Despite having geographic access, 44.6% of the participants reported that they had not used the SNAP-based nutrition incentive program in the past year. The overwhelming response as to why SNAP recipients had not tried the program was because they did not know it existed. However, when informed about the incentive program, most participants indicated that they were very likely to use it in the next six months. These findings highlight the importance of conducting dissemination research to explore how best to promote SNAP-based nutrition incentive programming to reach diverse groups by testing the impact of different messengers (e.g., trusted peers, nutrition educators, other influencers), channels of communications (e.g., word of mouth, social media, print), and messages. Our findings provide evidence for dissemination efforts that leverage social networks to raise awareness about SNAP-based incentive programs, an approach that has been effective in other research [31].

Additionally, our findings reinforce the importance of implementing SNAP-based incentive programs in multiple settings within a community to align with different food shopping routines of SNAP consumers. Much of the prior research on SNAP-based incentive programming focused on farmers’ market settings demonstrating several benefits, such as improvements in spending flexibility among households, fruit and vegetable purchasing and consumption, and opportunities for social connectivity [13,18,19,20,21]. In our study, incentive program users at farmers’ markets reported utilizing the program for the longest amount of time and were more likely to be older and White, which is consistent with research on farmers’ market shoppers in the U.S. [32]. Singleton and colleagues, for instance, found that non-Hispanic Black shoppers reported shopping at farmers’ markets less frequently and for fewer years than non-Hispanic Whites [33]. In our study, incentive program users at grocery stores were more likely to be Black and had lower levels of education (when compared to those who used at farmers’ markets or both). While implementation in grocery stores may promote racial equity in program reach, few studies have evaluated SNAP-based incentive programs within these settings [11,12]. Taken together, our findings highlight the need for new research to identify the right mixture of settings within a community for implementing SNAP-based incentive programming. Setting selection should be responsive to the shopping trends of local SNAP consumers with the broader goal of realizing the benefits of these programs more fully. Engaging SNAP consumers in the site selection process may increase the chance that the settings of implementation are a good fit for different segments of SNAP consumers.

Lastly, we found several segments of SNAP participants who may benefit from increased use of nutrition incentives to achieve programmatic goals of reducing disparities. The first segment included those experiencing food insecurity and other financial constraints related to low income. Despite having a high need to stretch food budgets, we found that the SNAP-based nutrition incentives were not widely used by this group. While lack of awareness about the program is a fundamental problem, these households may also have broader structural constraints or stressors influencing food purchasing that are not surmounted by the monetary value of a nutrition incentive. The second segment included those less confident about purchasing and preparing and those who less regularly consume fruits and vegetables. We found this segment was more likely to use incentives at grocery stores compared to farmers’ markets. The perceived value of the incentive program may be less for households not already integrating fruits and vegetables into their food routines. Reaching these two segments may require novel and synergistic strategies for program implementation and evaluation. Integrating SNAP-based incentive program implementation within broader efforts to reduce food insecurity, such as workforce development programming to promote job security or housing programs to promote stability, should be explored. In addition, findings point to the need for strategies that link culturally relevant nutrition education, skill building, and cooking classes with SNAP-based incentive program implementation to better reach those contemplating integration of fruits and vegetables into their food routines.

Our study has some limitations. The sample may not be reflective of SNAP consumers in non-urban settings, SNAP customers in other urban settings, those living in communities without any nutrition incentive programming, and non-English speaking populations. Additionally, the sample was mostly comprised of older women, whose perspectives may not reflect those of younger groups or men. The sample size was modest and thus reduced statistical power to rigorously compare some sub-groups of nutrition incentive program users. Findings were based on self-reports that could be subject to recall bias. We note that our study was designed to highlight the factors shaping nutrition incentive use choices, and we therefore restricted our attention to areas where a choice existed. As such, our study was designed to tease out determinants of choice rather than generate a representative assessment of SNAP-based incentive program use in the two cities included in the study. Finally, we used a cross-sectional study design, and while the reported associations generate critical hypotheses, they do not confirm causal relationships.

## 5. Conclusions

This study contributes key insights into the factors influencing use of SNAP-based nutrition incentive programs in urban contexts. Nutrition incentives represent a valuable resource to reduce cost barriers related to the purchasing of fruits and vegetables, yet our findings highlight that more efforts are needed to effectively raise awareness about these programs in communities. Overall, our research sheds light on the need for tailored implementation of incentive programming in diverse food retail settings that attends to different segments of SNAP consumers to promote equity in program reach and impact.

## Figures and Tables

**Figure 1 ijerph-19-04977-f001:**
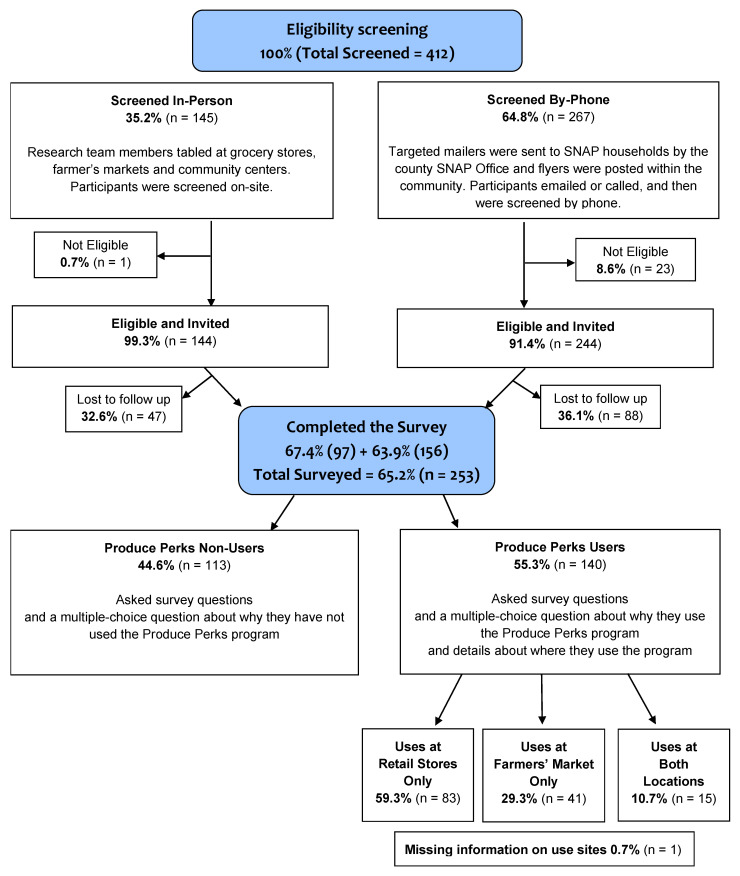
Study design and inclusion flow chart for the recruitment of snap consumers into a cross-sectional survey in 2019.

**Table 1 ijerph-19-04977-t001:** Characteristics of the cross-sectional survey participants: 253 SNAP recipients from Cleveland and Akron, Ohio, with access to nutrition incentive programming near their home in 2019.

Characteristics	n	%	Characteristics	n	%
**Gender**			**Number of children in the household**		
Female	206	81.4	0	171	67.6
Male	45	17.8	1	42	16.6
Other	1	0.4	≥2	40	15.8
Declined to respond	1	0.4	**Transportation for shopping**		
**Race**			Drive your own car	118	46.6
White	67	26.5	Depend on other options	135	53.4
Black	173	68.4	**Self-reported general health status**		
Other	13	5.1	Excellent	26	10.3
**Age in years**			Very good	47	18.6
<40	50	19.8	Good	81	32.0
40–59	96	37.9	Fair	78	30.8
60–79	98	38.7	Poor	21	8.3
>80	4	1.6	**Farmers’ market use in the past year**		
Missing	5	2.0	Yes	144	56.9
**Level of Education**			No	109	43.1
Some high school or less	23	9.1	**Length of time receiving SNAP (years)**		
High school graduate	105	41.5	<1	10	4.0
Some college or more	125	49.4	1–2	32	12.7
**Annual Household Income**			3–4	44	17.4
<10,000	87	34.4	≥5	166	65.6
≥10,000	166	65.6	Do not know/not sure	1	0.4
**Number of people in the household**			**Ever used Produce Perks**		
1	137	54.2	Yes	140	55.3
≥2	116	45.8	No	113	44.7

**Table 2 ijerph-19-04977-t002:** Participant characteristics that differ among nutrition incentive program users and non-users in Ohio, 2019.

	Nutrition Incentive Program Non-Users (n = 113)		Nutrition Incentive Program Users (n = 140)		*p*-Value for Difference
**Economic Domain**	**Median**	**IQR ^a^**	**Median**	**IQR ^a^**	
Annual income in dollars	10,000	8400, 13,600	12,000	9950, 18,000	<0.001 ^b^
Number of people supported by your SNAP	1	1, 2	1	1, 3	0.005 ^b^
Amount of SNAP money received last month	172	60.5, 193.5	192	110, 317.5	0.039 ^b^
**Spatial Temporal Domain**	**Median**	**IQR ^a^**	**Median**	**IQR ^a^**	
It is not convenient to buy fruits and vegetables (1 = strongly disagree, 5 = strongly agree)	2	1, 2	1	1, 2	0.005 ^b^
Distance to preferred store	3.3	1.4, 4.8	2.2	0.8, 4.1	0.020 ^b^
In the last year I obtained food from a:	**Frequency**	**%**	**Frequency**	**%**	
Farmers’ market	56	49.6	88	62.9	0.034 ^c^
Specialty store (e.g., ethnic store, bakery, meat market, seafood market, green grocer)	53	46.9	84	60.0	0.038 ^c^
Mobile pantry/free fruit and vegetable drop off	18	15.9	37	26.4	0.044 ^c^
How do you usually get to your food store/market/pantry					
Have a car	45	39.8	73	52.1	0.051 ^c^
Use someone else’s car	15	13.3	8	5.7	0.048 ^d^
**Social Domain**	**Median**	**IQR ^a^**	**Median**	**IQR ^a^**	
Think about friends, family, or people you know, about how many have used Produce Perks over the past 6 months? (1 = none, 5 = all)	1	1, 2	2	1, 3	<0.001 ^a^
Think about friends, family, or people you know, about how many have shopped at your farmers’ market over the past 6 months? (1 = none, 5 = all)	2 (Mean: 2.5)	2, 3	2 (Mean: 2.2)	2, 3	0.054 ^a^
**Personal Domain**	**Median**	**IQR ^a^**	**Median**	**IQR ^a^**	
Number of children in your household	0 (Mean: 0.5)	0, 1	0 (Mean: 0.8)	0, 1	0.034 ^a^
	**Frequency**	**%**	**Frequency**	**%**	
Food insecurity (USDA six-item food security score > 1, indicating low or very low food security)	70	62.0	69	49.3	0.044 ^c^
Moved in the last 12 months	22	19.5	15	10.7	0.050 ^c^
Education					
Some high school or less	15	13.3	8	5.7	0.061 ^b^
High school graduate	48	42.5	57	40.7	
Some college or more	50	44.3	75	53.6	
Total number of people in your household	1 (Mean:1.8)	1, 2	1 (Mean:2.2)	1, 3	0.086 ^b^

^a^ Interquartile range. ^b^ Wilcoxon–Mann–Whitney test (non-parametric analog of the *t*-test, assumes ordinal rather than normally distributed interval). ^c^ Chi-squared test. ^d^ Fisher’s exact test (non-parametric analog of the chi-squared test).

**Table 3 ijerph-19-04977-t003:** Characteristics that differ by nutrition incentive program setting of use: grocery store, farmers’ market, or both (at *p* < 0.1).

	Nutrition Incentive Program Users						
	Grocery Store Only (n = 83)		Farmers’ Market Only (n = 41)		Both (n = 15)		*p*-Value for Difference
	Median	IQR ^a^	Median	IQR ^a^	Median	IQR ^a^	
**Economic Domain**							
Over the past 6 months, about how much Produce Perks did you receive from the location of first use?	20	10, 50	20	10, 50	100	20, 200	0.030 ^b^
**Service Domain**							
How does the variety of fresh fruits and vegetables at your primary store compare to the variety of fresh fruits and vegetables at the farmers’ market? (1 = much greater variety, 5 = much less variety)	3	2, 4	4	3, 5	4	3, 5	<0.001 ^b^
**Spatial Temporal Domain**							
Number of visits per month to your primary store	4	2, 8	3	2, 4	2	3, 4	0.009 ^b^
**Social Domain**							
Think about friends, family, or people you know, about how many shopped at the promoted store offering Produce Perks over the past 6 months? (1 = none, 5 = all)	3	2, 5	2	2, 3	2.5	2, 4	0.054 ^b^
Think about friends, family, or people you know, about how many have shopped at your farmers’ market over the past 6 months? (1 = none, 5 = all)	2	2, 3	2	1, 3	2.5	2, 4	0.091 ^b^
**Personal Domain**							
Number of months since first Produce Perks use	3	1, 7	13	2, 30	8	3, 36	<0.001 ^b^
How often do you purchase fruits and vegetables when you go to your primary store? (1 = never, 5 = always)	4	3, 5	5	4, 5	5	5, 5	<0.001 ^b^
Number of times per month ate “other” vegetables (beyond beans, dark green or orange vegetables)	12	6, 28	20	8, 30	30	15, 30	0.007 ^b^
Confidence in preparing fresh green vegetables (1 = not at all confident, 5 = extremely confident)	4	4, 5	5	4, 5	5	5, 5	0.008 ^b^
Confidence in preparing root vegetables (1 = not at all confident, 5 = extremely confident)	4	4, 5	5	4, 5	5	4, 5	0.018 ^b^
Education	**Frequency**	%	**Frequency**	**%**	**Frequency**	**%**	0.025 ^b^
Some high school or less	8	9.6	0	0.0	0	0.0	
High school graduate	38	45.8	16	38.1	3	20.0	
Some college or more	37	44.6	26	61.9	12	80.0	
	**Median**	**IQR ^a^**	**Median**	**IQR ^a^**	**Median**	**IQR ^a^**	
Age in years	56	37, 62	60	47, 69	53	42, 57	0.038 ^b^
	**Frequency**	**%**	**Frequency**	**%**	**Frequency**	**%**	
Black, race	63	75.9	23	54.8	10	66.7	0.056 ^c^
	**Median**	**IQR ^a^**	**Median**	**IQR ^a^**	**Median**	**IQR ^a^**	
Number of times per month ate beans	3	2, 4	4	2, 5	4	3, 8	0.062 ^b^
Number of adults in your household	1 (Mean:1.3)	1, 1	1 (Mean:1.6)	1, 2	1 (Mean:1.6)	1, 2	0.067 ^b^
Number of times per month cooked meals made from scratch or using whole foods	16	8, 30	20	12, 30	28	20, 30	0.076 ^b^

^a^ Interquartile range. ^b^ Kruskal–Wallis (non-parametric analog of ANOVA, assumes ordinal rather than normally distributed interval). ^c^ Fisher’s exact test (non-parametric analog of the chi-squared test).

## Data Availability

The data include addresses and will not be publicly posted. Inquiries about the data and requests for access can be sent to Darcy A. Freedman (daf96@case.edu).

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
