# Peer review of "Understanding SNAP Recipient Characteristics to Guide Equitable Expansion of Nutrition Incentive Programs in Diverse Food Retail Settings"

_ijerph, 2022, doi:10.3390/ijerph19094977_

Round 1

Reviewer 1 Report

 Thank you for the opportunity to review this manuscript which focuses on the understanding SNAP recipient characteristics. However, I have some  major comments about the review below:

Abstract

  1. Line 12-13 “Nutrition incentive programs seek to decrease food prices and increase healthy food 12 purchasing among people receiving Supplemental Nutrition Assistance Program (SNAP)”.

Comment: I do not think that the Nutrition incentive programs decreases food prices and increase healthy food. Rather, it increases the access to adequate food  and affordability of healthy food option or encourage consumption of health food options through various strategies.

  1. Line 13-16: Evaluation of incentive programming has focused on discrete settings, such as farmers’ markets or grocery  We took a holistic view by examining use patterns, characteristics, and preferences among  253 SNAP consumers with access to nutrition incentive programming at both a farmers’ market and  grocery store located within five miles of their home.

Comment: I am not sure what is different in your research when compared to what has been done as you indicated that  “Evaluation of incentive programming has focused on discrete settings, such as farmers’ markets or grocery 14 stores” which is similar to  the current study. The knowledge gap or new information that the current study is/are providing is/are not clear.

  1. A major finding in the study was the lack of awareness of the program and this is not mentioned at all in the abstract.

 Introduction

  1. Line 48-49: Evaluations of nutrition incentive programming found they increase access to and consumption of a wider variety of fruits and vegetables among those who use the program and are associated with reductions in food insecurity and improvements in diet quality among people receiving SNAP [11,12] .

Comment:  It is not clear if the authors are referring to the GusNIP  or  a variety of nutrition incentive programs in this sentence.

  1. However, reach of the program is limited in many communities, and more implementation research is needed to increase the 51 number of people who are benefiting from the program [13].

Comment: “the” should be before the word reach i.e. it should be  the reach of the program

  1. Line 72-79: “Most extant research is focused on the potential impacts of offering a nutrition incentive program at either farmers’ markets [12,19–21] or grocery stores [11,17] as independent program operations. In this research, we extend existing studies by taking a holistic view 74 of nutrition incentive programming. The aim of our cross-sectional study was to examine use patterns, characteristics, and preferences among SNAP recipients with access to nutrition incentive programming at both a farmers’ market and a grocery store (located within five miles of their home). Findings will provide insights for reducing diet-related  disparities through the development of nutrition incentive strategies that are population- specific.

Comment. The knowledge gap or new information that the current study is/are providing is/are not clear.

  1. Overall comment on introduction:  Although the topic of the study is very interesting, the introduction lacks adequate content and  It seems to touch on various issues SNAP, nutrition incentive,  access to fruits and vegetables and farmers market and grocery stores without a clear evidence of the gap or new knowledge the study aims to show. As such, I suggest a the introduction be written to include concise, relevant and  information that clearly showcase the reason/importance of  the aim of the study.
  2. Line 91: 92: It is not clear why  part of the requirement for eligibility was “for  the grocery store and farmers’ market were located within three miles of one another”
  3. Line 116: Why is one of the requirements for participating is the study must be “English speaking “as some of the most resource-poor might be non- English speakers or not fluent in English and hence this is a limitation of the study.

Data Analysis

  1. Line 165: “Kruskal Wallace”. Please check the spelling.  This should be Kruskal Wallis test.

  Results and Discussion

  1. Overall comments: This discussion section has to be a bit more concise with interpretation of the results for readers  and its significance of the finding and not be a repetition of results e.g  Line 270-286 is repetitive. It also did not provide significance of those using the program being more food secure.
  2. Line 203 : Produce Perk User Profile: I think the user profile of this group of people should be fleshed our more i.e. the fact that they had more children (which I think is a key point). Of note, more  than two- thirds of the participants  did not have children ( Table 1).  Overall, the strongest statistical findings and likely predictors of Produce Perks use were having a higher income and knowing more people who use the incentive program.  If a key factor is lack of awareness then I think there should be suggestion of  more research on how the awareness of the program is being done currently and how this can be expanded to  suit all participant types
  3. Line 262-266: Three key lessons learned from our research include: (1) multi-pronged approaches are needed to engage diverse audiences of SNAP recipients in nutrition incentive programming; (2) those who may most benefit from additional food assistance benefits through nutrition incentives were least likely to use the program; and  (3) insufficient awareness of nutrition incentive programming was a critical yet modifiable impediment to its use.

Comment:  suggested solutions or recommendation for the  changes would have been beneficial for this study. E,g how do you solve the problem of those needing the nutrition incentives were less likely to use the program.

  1. Line 239-240: They also reported that they ate beans (p=0.062) and  other vegetables (i.e., beyond beans, dark green or orange vegetables) less frequently 240 (p=0.007).

Comments: It can be suggested that a factor to lower use of the program could be  low preference to consumption of fruits and veg compared to other food types, and lack of  knowledge on different ways in which vegetables can be cooked  and this could further explored in further studies. So I could be that the use of  program could be increased by maybe more awareness /appreciation of the need or importance to consume vegetables and the different ways to prepare them.

  1. Line 262-266: Three key lessons learned from our research include: (1) multi-pronged approaches are needed to engage diverse audiences of SNAP recipients in nutrition incentive programming; (2) those who may most benefit from additional food assistance benefits through nutrition incentives were least likely to use the program; and  (3) insufficient awareness of nutrition incentive programming was a critical yet modifiable impediment to its use.

Comment: suggested solutions or recommendation for the  changes would have been beneficial for this study. E,g how do you solve the problem of those needing the nutrition incentives but we were less likely to use the program.

  1. Line 292-296: In contrast, we found that SNAP participants with the greatest need to stretch food budgets (i.e., those with low food security) were the least likely to use the nutrition incentive program. These households may have broader structural constraints on food purchasing that are not surmounted by the value of a nutrition incentive. For instance, cost savings through an incentive may not be sufficient to address financial and time costs related to securing transportation to go to a farmers’ market or grocery store offering a nutrition  incentive program.

Comments:  I do not follow this rationale.  I would assume that their being food insecure might be linked to not being a user of the program as even though the non-users had a lower income, most were not even aware of the program, but are willing to participate once they were well- informed. Of note, only 3.5% indicated that transportation  and (2.7%) high cost were barriers.

  1. Line 322:  “lower educational attainment” compared to the Whites or ?
  2. Line 344-348: The limitation section should be more extensive e.g I would think that having only English speaking participants is a limitation and the fact that maybe it only cover those within  certain radius of the farmers’ market or grocery stores and so it is also not a reflect Akron and Cleveland 

Conclusion

  1. Line 353-356: Our research sheds light on the need for tailored implementation of incentive programming that at tends to diverse segments of SNAP consumers to promote equity in program reach in 355 grocery retail and farmers’ market settings.

Comments: The result of the study does not state there is  challenge in the implementation but more in the lack of awareness of the program  and the lack of information regarding the program  etc

 Others:

Supplementary material:  I am not sure of the relevance of the supplementary material which is not clear to me as it provides just the values .Also, it seems  like the analysis  was conducted  between (Nutrition Incentive Program  Users  Vs. Non-Users)  and (Nutrition Incentive Program by Location of Use Retail vs. Farmer’s Market vs. Both ) - which  is not really possible.

Reviewer 2 Report

Thank you for the opportunity to review this manuscript. Research related to increasing equity within nutrition assistance is as difficult as it is necessary. 

Broadly the majority of the manuscript is very well-organized and connected throughout with only a few considerations for the authors. See comments below. 

Abstract

Line 16 - what defines access to NIP?

Introduction

Line 31 - Recommend reviewing and updating on literature relevant to FI and disparities that are a result of inequities and the exacerbation as a result of the pandemic. There is research indicating FI did increase after the pandemic and especially in historically marginalized populations.

Line 78 implies the findings are generalizable to similar populations in order to improve equitable access of NIP's, however your study is specific to a single region of the U.S. and you state this in the discussion as a limitation. I recommend adding an identifier like 'strategies that are specific to the unique urban populations in Ohio' or similar. The sample of your population is not like other populations who experience nutrition and health disparities due to regional and historical differences in racial, ethnic, or economic disparities and your study is bound by those near NIPs, which inherently do not serve rural populations.

Methods

Line 88 - Not clear why these eligibility criteria were chosen and what is the rationale for not including non-participating sites for comparison or insight into how Produce Perks can reach those they don't currently serve?

Line 142 - Would connect this framework to the introduction as your methods, findings, and discussion are informed by this framework.

Line 148 - Did you ask a question whether their preferred store participated in NIP? It is possible that store preference changed as a function of a store nearby accepting/participating in Produce Perks? If so I would indicate in discussion

Line 166 - If this is exploratory in nature, it would be of interest to include the use of the qualitative methods (collection and analysis) in this manuscript to better support the generation of future hypotheses and theory testing (Nutritious food access model). 

Results

Line 200 - There was no mention of collecting open-ended or qualitative responses in the methods section. Update methods section and where appropriate in tables or supplemental table and discussion section. It should be noted in the limitations that you did not assess non-participants in what would motivate them to participate in the program.

Table 2 - First variable in spatial domain. Median for NIP users is lower than non-users indicating users perceive it is not convenient to buy FV more than non-users. Is this correct? Your narrative indicates otherwise. Please clarify or update table or narrative accordingly.

Discussion

Line 270 - Your study highlights that access is not similar. Would re-word to 'Despited all participants being within 5 miles of a FM or store that provided produce perks, 44.6%...'

Line 272 - Cite related research.

As noted in data analysis section, it is not clear what the association is between living closer to preferred store and program use unless the preferred store participated in Produce Perks. Clarify when describing measures and expand on here as to whether your study is able to discern this.

-No mention of qualitative data or findings related to FV intake as you did not find a difference in FV intake between users and non-users, only differences as a function of shopping location

Line 287 - What factors or characteristics influence this and is there research in support of this or is this new/unique to our understanding?

Line 291- 

Line 291 - This seems like a restatement of the above paragraph and your findings. Would consider deleting or re-wording/merging with Line 293-294. Example- 'In contrast our findings indicate SNAP households that were less likely to participate in NIP may have broader structural constraints...'

Line 297- Any specific examples you can provide based on hwo to make SANP and NIP more inclusive? It seems as though these folks may need more SNAP benefits and/or higher incentives to offset economic challenges and higher access to nutrition education and promotion programs. Policies to support low-income households who are displaced or moving frequently?

Line 303- Again, these are broad nor rooted in evidence from other nutrition incentive programs (double bucks, produce prescriptions) that have cited specific approaches and methods to support these considerations.

Line 324- There is also research indicating these populations are interested in local food systems that they are familiar with and support/uplift their communities, however community-engaged approaches, as mentioned above, are needed in order to develop NIP that use local food systems that these populations and communities are more involved in and/or benefit from. Discuss the nuances of this challenge against NIP wanting to prop up local food systems vs. moving towards equitable access. 

Reviewer 3 Report

Respected Authors,

Thank you very much for the interesting reading. You manuscript is prepared very well and brings interesting information about the nutrition incentive program in the Ohio with the shopping behavior of recipients and their characteristics.

In the introduction is described current problematics with SNAP, which provides and explains the mechanism and history of nutrition incentive programs. 

M&M is described very well in the details, however there was used questionnaire for the data collection and the template with the questions (it could improve the repeatability of the experiment and number of citations in the future).

I appreciate that there were used so many statistical analysis methods - I think that authors used all the possibilities for this type of data and research.

Results are represented in the tables and are presented very well. I think that there could be presented another results of experiment - the correlation between the health status of respondents and the produce perks users - my expectation is that farmers perks users will be in better health conditions. If yes - the results of the experiment could support the local farmers for example.

The discussion is also prepared very well. There is high number of literature sources from the last 5 years (18) and old resources are used in the minimal numbers.

Then, in the conclusion answer on the hypothesis which you provide in the aims (the last part of the introduction). There is missing the answer on the characteristics and preferences among SNAP recipients. For example in the abstract are the answers on the hypothesis answered.

In general, the manuscript is prepared very well and I hope that my small recommendation help to improve your manuscript despite its high quality.

Best Regards,

Reviewer.
